# *ScatterSample*: Diversified Label Sampling for Data Efficient Graph Neural Network Learning

**Zhenwei Dai**
Department of Statistics
Rice University

**Vasileios Ioannidis**
Amazon Web Services

**Soji Adeshina**
Amazon Web Services

**Zak Jost**
Amazon Web Services

**Christos Faloutsos**
Department of Computer Science
Carnegie Mellon University

**George Karypis**
Amazon Web Services

## Abstract

What target labels are most effective for graph neural network (GNN) training? In some applications where GNNs excel-like drug design or fraud detection, labeling new instances is expensive. We develop a data-efficient active sampling framework, ScatterSample, to train GNNs under an active learning setting. ScatterSample employs a sampling module termed DiverseUncertainty to collect instances with large uncertainty from different regions of the sample space for labeling. To ensure diversification of the selected nodes, DiverseUncertainty clusters the high uncertainty nodes and selects the representative nodes from each cluster. Our ScatterSample algorithm is further supported by rigorous theoretical analysis demonstrating its advantage compared to standard active sampling methods that aim to simply maximize the uncertainty and not diversify the samples. In particular, we show that ScatterSample is able to efficiently reduce the model uncertainty over the whole sample space. Our experiments on five datasets show that ScatterSample significantly outperforms the other GNN active learning baselines, specifically it reduces the sampling cost by up to **50**% while achieving the same test accuracy.

## 1 Introduction

How to spot the most effective labeled nodes for GNN training? Graph neural networks (GNN) [KW16; Vel+17; Wu+19a] which employ non-linear and parameterized feature propagation [ZG02] to compute graph representations, have been widely employed in a broad range of learning tasks and achieved state-of-art-performance in node classification, link prediction and graph classification. Training GNNs for node classification in the supervised learning setup typically requires a large number of labeled examples such that the GNN can learn from diverse node features and node connectivity patterns. However, labeling costs can be expensive which inhibits the possibility of acquiring a large number of node labels. For example, the GNNs can be used to assist the drug design. However, evaluating the properties of a molecule is time consuming. It usually takes one to two weeks for evaluation using the current simulation tools, not to mention the cost spent on the laboratory experiments.

Active learning (AL) aims at maximizing the generalization performance under a constrained labeling budget [Set09]. AL algorithms choose which training instances to use as labeled targets to maximize the performance of the learned model. Previous research in AL algorithms for GNN training can be categorized with respect to whether the AL methods take into account the model weights (model aware) or can be applied to any model (model agnostic). Model agnostic algorithms label a represen-

, *ScatterSample*: Diversified Label Sampling for Data Efficient Graph Neural Network Learning. *Proceedings of the First Learning on Graphs Conference (LoG 2022)*, PMLR 180, Virtual Event, December 9–12, 2022.

tative subset of the nodes such that the labeled nodes can cover the whole sample space [Wu+19b; Zha+21]. Model aware AL algorithms leverage the GNN model to compute the node uncertainty, which combines both the input features and graph structure [CZC17; Gao+18]. AL then picks the nodes with the highest uncertainty.

However, maximizing the uncertainty of the labeled nodes may not balance the exploration and exploitation of the classification boundary [KVAG19]. For example, if there exist a group of nodes close to the classification boundary but are clustered in a small region of the graph, just labeling the most uncertain nodes could only explore that specific region of the classification boundary, while others are ignored, and the classification boundary is not well explored. Thus, our first main contribution is to simultaneously consider the node uncertainty and the diversification of the uncertain nodes over the sample space.

**Challenges of diversifying uncertain nodes.** Graph data present additional challenges to diversify the uncertain nodes. Diversification requires modeling the sample space using carefully selected representations for the nodes. However, there are two challenges of a suitable node representations.

Challenge 1: Sample space for graph data requires a representation which takes both the graph structure and node features into account (see section sec 4.2).

Challenge 2: The representation should be robust to the model trained so far, and not be biased by the limited amount of available labels.

**Our approach.** We develop *ScatterSample* for data-efficient GNN learning. *ScatterSample* allows us to explore the classification boundary while exploiting the nodes with the highest uncertainty. To diversify the uncertain samples on graph-structured data, *ScatterSample* includes a *DiverseUncertainty* module to address the two challenges above, which clusters the uncertain nodes representations over the whole sample space.

**Our Contributions.** The contributions of our work are the following.

- **Insight:** *ScatterSample* is the first method that proposes and implements diversification of the uncertain samples for data efficient GNN learning.
- **Effectiveness:** We evaluate *ScatterSample* on five different graph datasets, where *ScatterSample* saves up to **50%** labeling cost, while still achieving the same test accuracy with state-of-the-art baselines.
- **Theoretical Guarantees**: Our theoretical analysis proves the superiority of *ScatterSample* over the standard, uncertainty-sampling method (see Theorem 5.1). Simulation results further confirm our theory.

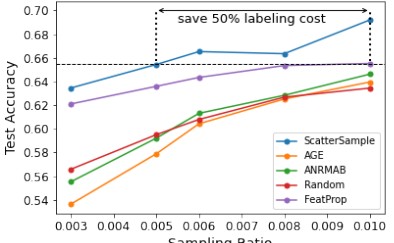

**Figure 1:** *ScatterSample* **wins:** test accuracy vs. sampling ratio on the ogbn-products dataset (62M edges).

## 2 Related Work

This section will review the uncertainty based active learning research and implementation of active learning in GNNs.

**Active Learning (AL):.** Active learning aims at selecting a subset of training data as labeling targets such that the model performance is optimized [Set09; Han+14]. Uncertainty sampling is one major approach of active learning, which labels a group of samples to maximally reduce the model uncertainty. To achieve this goal, uncertainty sampling selects samples around the decision boundary [THTS05]. Uncertainty sampling has also been applied to the deep learning field, and researchers have proposed different methods to measure the uncertainty of samples. For example, Ducoffe and Precioso [DP18] developed a margin based method which uses the distance from a sample to its smallest adversarial sample to approximate the distance to the decision boundary.

**AL and GNNs:** AL with GNNs requires to consider the graph structure information into the node selection. Wu et al. [Wu+19b] uses the propagated features followed by K-Medoids clustering of nodes to select a group of representative instances. Zhang et al. [Zha+21] measures importance of nodes through combining the diversity and influence scores. However the above approaches do not account for the learned GNN model, which may limit the generalization performance. Uncertainty sampling has also been implemented to select nodes. Cai et al. [CZC17] propose to use a weighted

average of the node uncertainty, graph centrality and information density scores. Gao et al. [Gao+18] further propose a different approach to combine the three features with multi-armed bandit techniques. Although useful, these approaches aim choose nodes with the highest uncertainty and may be challenged if the selected nodes are clustered in a small region of the graph, which will not provide good graph coverage. Our work addresses this limitation by diversifying the selected nodes based on the graph structure.

## 3 Preliminaries

**Problem Statement .** Given a graph $\mathcal{G} = (\mathcal{V}, \mathcal{E})$, where $\mathcal{V}$ is the set of nodes with $N = |\mathcal{V}|$ nodes and $\mathcal{E}$ is the set of edges. The set of nodes is divided into the training set $\mathcal{V}_{train}$, validation set $\mathcal{V}_{valid}$ and testing set $\mathcal{V}_{test}$. Each node $v_n \in \mathcal{V}$ is associated with a feature vector $\mathbf{x}_n \in \mathbb{R}^d$ and a label $y_n \in \{1, 2, \dots, C\}$. Let $\mathbf{X} \in \mathbb{R}^{N \times d}$ be the feature matrix of all the nodes in the graph, where the $i$-th row of $\mathbf{X}$ corresponds to $v_n$, $\mathbf{y} = (y_1, y_2, \dots, y_n) \in \mathbb{R}^n$ is the vector containing all the labels. To learn the labels of the nodes, we train a GNN model $M$ which maps the graph $\mathcal{G}$ and $\mathbf{X}$ to the the prediction of labels $\hat{\mathbf{y}}$.

**Active Learning:** Active learning picks a subset of nodes $S \subset \mathcal{V}_{train}$ from the training set and query their labels $\mathbf{y}_S$. A GNN model $M_S$ is trained with respect to the feature matrix $\mathbf{X}$ and $\mathbf{y}_S$. Given the sampling budget $B$, the goal of active learning is to find a set $S$ ($|S| \leq B$) such that the generalization loss is minimized, i.e.

$$\underset{S:|S|\leq b}{\arg \min} \, \mathbb{E}_{v_n \in \mathcal{V}_{test}} \left( \ell(y_n, f(\mathbf{x}_n | \mathcal{G}, M_S)) \right).$$

### 3.1 Graph neural networks and message passing

In this section we present the basic operation of the GNN at layer $l$. With the message passing paradigm, the GNN layer updates for most GNN models can be interpreted as message vectors that are exchanged among neighbors over the edges and nodes in the graph.

For the following let $\mathbf{h}_v^{(l)} \in \mathbb{R}^{d_1}$ be the hidden representation for node $v$ and layer $l$. Consider $\phi$ that is a message function combining the hidden representations for nodes $v, u$. Next, using the message vectors for neighboring edges the node representations are updated as follows

$$\mathbf{h}_v^{(l+1)} = \psi \left( \mathbf{h}_v^{(l)}, \rho(\{\phi(\mathbf{h}_v^{(l)}, \mathbf{h}_u^{(l)}) : (u, v) \in \mathcal{E}\}) \right) \tag{1}$$

where $\rho$ is a reduce function used to aggregate the messages coming from the neighbors of $v$ and $\psi$ is an update function defined on each node to update the hidden node representation for layer $l + 1$. By defining $\phi, \rho, \psi$ different GNN models can be instantiated [KW16; DBV16; Bro+17; IMG20]. These functions are also parameterized by learnable matrices that are updated during training.

## 4 Proposed method: *ScatterSample*

We propose the *ScatterSample* algorithm, which dynamically samples a set of diverse nodes with large uncertainties in order to more efficiently explore the classification boundary during GNN training. At each round, our method calculates the uncertainty for all nodes with the GNN model trained so far. Then, *ScatterSample* clusters the top uncertain nodes and selecting nodes from each cluster to obtain diverse samples. The labels of the selected nodes are queried and used as supervision to continue training the GNN model for the next round. This section explains our method in detail.

### 4.1 Selecting the uncertain nodes

The uncertainty of a node is measured by the information entropy. Given a trained GNN model at the $t$-th sampling round, *ScatterSample* first computes the information entropy $\phi_{entropy}(v_n)$ of nodes in $\mathcal{V}_{train}$ based on the current GNN model, i.e.

$$\phi_{entropy}(v_n) = -\sum_{j=1}^{C} \log(\mathrm{P}\left[Y_n = j \mid \mathcal{G}, \mathbf{X}, M\right])\mathrm{P}\left[Y_n = j \mid \mathcal{G}, \mathbf{X}, M\right] \tag{2}$$

where $\mathrm{P}\left[Y_n = j \mid \mathcal{G}, \mathbf{X}, M\right]$ is probability that node $v_n$ belongs to class $j$ given the GNN model $M$. Then, *ScatterSample* ranks all the nodes in order of decreasing uncertainty, and picks the ones with the largest information entropy into a candidate set $\mathcal{C}_t \subset \mathcal{V}_{train}$. Different than traditional AL techniques that select training targets solely based on uncertainty, we then move on to pick a diverse subset of the uncertain nodes over the sampling space.

## 4.2  Diversifying uncertain nodes

Our goal is to ensure the diversity of selected nodes for labeling, by exploring the node distribution over the sample space. At this point naturally, the question arises *How to model the sample space?* We need a representation for nodes to define the space, based on which we could measure the samples' distances. A straightforward approach is to use the GNN embedding space since the classification boundary is directly depicted there. However, GNN embeddings fail to address the two challenges in the introduction section.

First, with active learning, a limited number of labeled nodes are available in the initial stages. Hence, only the already labeled nodes may have reliable GNN embeddings and biased subsequent samples. Second, GNN embeddings for node classification may not carry enough information for diversification. GNNs usually do not have an MLP layer connecting to the output. The final GNN outputs of uncertain nodes are not diverse enough since the high uncertain nodes may have similar class probabilities (class probabilities close to uniform). Conversely, embeddings of intermediate GNN layers may have an appropriate dimension but lack information of the expanded ego-network.

These drawbacks are confirmed in Sec. 6.2, where we show that using GNN embeddings as proxy representations leads to a performance drop. Moreover, different from other machine learning problems, the nodes are correlated with each other, and we also need to take the graph structure into account when diversifying the samples. Hence, to address all these considerations we will employ a $k$-step propagation of the original node features based on the graph structure as a proxy representation for the nodes. The $k$-step propagation of nodes $\mathbf{X}^{(k)} = (\mathbf{x}_1^{(k)}, \mathbf{x}_2^{(k)}, \ldots, \mathbf{x}_N^{(k)})$ is defined as follows

$$\mathbf{X}^{(k)} := \mathbf{S}\mathbf{X}^{(k-1)} \tag{3}$$

where $\mathbf{S}$ is the normalized adjacency matrix, and $\mathbf{X}^{(0)}$ are the initial node features. The operation in (3) is efficient and amenable to a mini-batch implementation. Such representations are well-known to succinctly encode the node feature distribution and graph structure. Next, we calculate the proxy representations for the candidate high uncertainty nodes in the set $\mathcal{C}_t$. To maximize the diversity of the samples, we cluster the proxy representations in $\mathcal{C}_t$ using $k$-means++ into $B_t$ clusters [AV06], and select the nodes closest to the cluster centers for labeling, by using the $L_2$ distance metric. One node from each cluster is selected that amounts to $B_t$ samples.

---

**Algorithm 1** *ScatterSample* Algorithm

---

1: **Input**: $\mathcal{V}_{train}$, GNN model $M$, number of propagation layers $k$, number of sampling round $T$, sampling redundancy $r$, initial sampling budget $B_0$ and total sampling budget $B$.
2: Initialize $S = \emptyset$
3: Compute $\mathbf{x}_n^{(k)} \ \forall n \in \mathcal{V}_{train}$ as in (3).

4: **Initial Sampling:**
5: Use $k$-means++ to cluster $\{\mathbf{x}_n^{(k)}\}$ into $B_0$ clusters.
6: Add a node closest to the cluster center per cluster to $S$.
7: Query the labels of nodes $v_n \in S$, denoted by $\mathbf{y}_S$.
8: Train model $M$ using $(\mathbf{y}_S, \mathbf{X}, \mathcal{G})$.

9: **Dynamic Sampling:**
10: Initialize sampling round $t = 1$
11: **while** $t < T$ **do**
12:    Let $B_t = \min(B - |S|, (B - B_0)/T)$
13:    Use the **DiverseUncertainty** algorithm to select $S_t$
14:    Query the labels of $S_t$, and update $S = S \cup S_t$.
15:    Train model $M$ over $(\mathbf{y}_S, \mathbf{X}, \mathcal{G})$. Update $t = t + 1$.
16: **end while**

---

Clearly, the size of the candidate set $|\mathcal{C}_t| \geq B_t$, however deciding how many candidate nodes to choose from is important. We parameterize the size as a multiple of the selected nodes namely $|\mathcal{C}_t| = rB_t$, where $r > 1$ is the sampling redundancy. If $r$ is too small, the selected nodes are closer to the classification boundary (have larger information entropy) but the nodes selected may not be diverse enough. On the other hand, if $r$ is too large, the set will be diverse, but the selected nodes may be far away from the classification boundary. Therefore, it is critical to pick a suitable $r$ to achieve

a sweet point between diversity and uncertainty. We leave the discussion of choosing $r$ to Sec. 6.2. Besides empirical validation with experiments in five real datasets (see Sec. 6), our diversification approach is theoretically motivated (see Sec. 5).

---

**Algorithm 2** DiverseUncertainty Algorithm

---

1: **Input**: $\mathcal{V}_{train}$, $\{\mathbf{x}_n^{(k)} \ \forall n \in \mathcal{C}_t\}$, $r$, $B_t$
2: Compute $\phi_{entropy}(v) \ \forall v \in \mathcal{V}_{train}$; see 2).
3: $\mathcal{C}_t \leftarrow \{rB_t$ nodes with largest $\phi_{entropy}(v)\}$.
4: Use $k$-means++ to cluster the $\mathbf{x}_n^{(k)}$ (for all $n \in \mathcal{C}_t$) into $B_t$ clusters.
5: $S_t \leftarrow \emptyset$
6: **for** $j = 1, 2, \ldots, B_t$ **do**
7:     Compute the cluster center $\mathbf{v}_j$ of cluster $j$
8:     Pick node $x \leftarrow \arg\min_{n \in \mathcal{C}_t} \left\| \mathbf{x}_n^{(k)} - \mathbf{v}_j \right\|$
9:     $S_t \leftarrow S_t \cup \{x\}$
10: **end for**
11: Return $S_t$

---

The pseudo code of *ScatterSample* is shown in Algorithm 1. *ScatterSample* is a multiple rounds sampling scheme, which includes an initial sampling step and dynamic sampling steps. *ScatterSample* first computes the $k$-step features propagation of all the nodes in the training set using (3), and clusters them into $B_0$ clusters, where $B_0$ is the initial sampling budget. Then, *ScatterSample* picks the nodes closest to the cluster centers as the initial training samples and queries their labels. The purpose of clustering $k$-step feature propagations is to enforce the initial training set to spread out over the whole sample space. It is also helpful to explore the classification boundary since if the initial sampled nodes are not diverse enough, we cannot picture the classification boundary of the regions that are far away from the initial training samples. *ScatterSample* repeats the dynamic sampling described in Algorithm 2 until the sampling budget $B$ is exhausted. The next section fortifies our diversification method with theoretical guarantees.

## 5 Theoretical analysis

In Sec. 6.2, we have shown that DiverseUncertainty is significantly better than Uncertainty algorithm. In this section, we provide theoretical analysis and simulation results to demonstrate the benefits of DiverseUncertainty and explains why MaxUncertainty algorithm may fail. The results presented here give a theoretical basis for the superiority of our method as established in the experiments in Section 6.

### 5.1 Analysis setup

For the analysis, we employ the Gaussian Process (GP) model [O'H78]. GP models offer a flexible approach to model complex functions and are robust to small sample sizes [See04]. Moreover, the uncertainty of the prediction can be easily computed using a GP model. Neural network models and GNNs interpolate the observed samples, while GPs provide a robust framework to interpolate samples, that is amenable to analysis.

Assume the label $y_i \in \mathbb{R}$ is dependent on the propagated features $\mathbf{x}_i^{(k)}$ through a GP model. The label $y_i$ is modeled by a Gaussian Process, where $(\mathbf{y} \mid \mathbf{X}^{(k)}) \sim N(\mathbf{1}\mu, \mathbf{K}(\mathbf{X}^{(k)}))$ and $\mathbf{K}(\mathbf{X}^{(k)})$ is the Gaussian kernel matrix. The kernel is parameterized by $\mathbf{K}_{ij}(\mathbf{X}^{(k)}) = K(\mathbf{x}_i^{(k)}, \mathbf{x}_j^{(k)}) = \exp\left(-\frac{1}{2}(\mathbf{x}_i^{(k)} - \mathbf{x}_j^{(k)})^T \Sigma^{-1}(\mathbf{x}_i^{(k)} - \mathbf{x}_j^{(k)})\right)$, where $\Sigma = diag(\theta_1, \theta_2, \ldots, \theta_d)$. Consider that the sample space of $\mathbf{x}^{(k)}$ can be clustered into $m$ clusters $\mathcal{S}_1, \mathcal{S}_2, \ldots, \mathcal{S}_m$, and denote the cluster centers as $\mathbf{c}_1, \mathbf{c}_2, \ldots, \mathbf{c}_m$. Without loss of generality, denote the radius of the cluster, $d_1 \leq d_2 \leq d_3 \leq \cdots < d_m$. The clusters are well separated and the distance between the cluster centers are larger than $\delta$, i.e. $\min_{i \neq j} \left\| \mathbf{c}_i - \mathbf{c}_j \right\|_2 \geq \delta$ ($\delta > 2d_m$). Moreover, we consider that there does not exist a cluster dominating the sample space, $d_m^2 \leq \tau \sum_{j=1}^{m-1} d_j^2$ and the samples are uniformly distributed over the clusters.

## 5.2 MaxUncertainty vs DiverseUncertainty

Here, we show that DiverseUncertainty could significantly achieves smaller mean squared error (MSE) compared to MaxUncertainty. Without loss of generality we consider $m$ clusters and the following definitions.

- **MaxUncertainty** Select $2m$ most uncertain samples.

- **DiverseUncertainty** Select the 2 most uncertain samples from each cluster.

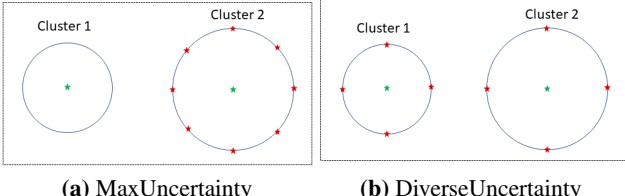

**(a)** MaxUncertainty      **(b)** DiverseUncertainty

**Figure 2:** The area enclosed by the blue circles is the sample space of propagated features (2D case). The green stars are sampled nodes during initial sampling (cluster center). The red stars are the sampled nodes during uncertainty sampling. (a) MaxUncertainty picks the nodes with largest uncertainty, which is equivalent to sampling the boundary of cluster 2. (b) DiverseUncertainty diversifies the clustered nodes, and samples the boundary of both clusters.

Before presenting the theory we illustrate the operation of our method and **MaxUncertainty** in Figure 2. *ScatterSample* first clusters the samples on the propagated feature space (blue circles in Figure 2), and selects the nodes closest to the cluster centers for initial training (green stars in Figure 2). Then, during the dynamic sampling steps, we compute the uncertainty using equation 4. MaxUncertainty approach will select the nodes with the largest uncertainty. Under our setup, it is equivalent to sample nodes at the boundary of the largest cluster since the distance to the cluster center is the most important factor of uncertainty (Figure 2(a)). While DiverseUncertainty will diversify the high uncertainty nodes, which is equivalent to sample from the boundary of each cluster (Figure 2(b)). The red stars of Figure 2 show the nodes labeled during the uncertainty sampling stage. Since MaxUncertainty algorithm only labels the nodes in cluster 2, cluster 1 is ignored the prediction uncertainty of cluster 2 cannot be reduced. On the contrary, DiverseUncertainty samples nodes from both cluster 1 and 2. Thus, it could reduce the prediction uncertainty in both clusters.

Then, the following theorem quantifies the relationship of the MSEs of both algorithms under the setup of Sec. 5.1.

**Theorem 5.1.** *Consider a case where feature dimension* $d = 1$. *With the above notation and assumptions, let* $r_i = \exp\left[-\frac{d_i^2}{2\theta}\right]$. *If we satisfy* $d_m^2 \geq d_{m-1}^2 + 4\log\theta$ *and* $\delta \geq d_m + \max\left(\sqrt{d_m^2 + \theta\log(9m)}, 2\theta\log(\frac{3\sqrt{m}}{1-r_m})\right)$, *we have*

$$\frac{MSE(f(x)|MaxUncertainty)}{MSE(f(x)|DiverseUncertainty)} \geq \frac{1}{2(1+\tau)}\frac{1+r_m^2}{1-r_m} - \frac{8}{3} = \frac{1}{\tau+1}O(\theta).$$

*Proof: The complete proof is included in Appendix B.*

Theorem 5.1 suggests that when the GP function is smooth enough (large $\theta$), the MaxUncertainty will have larger MSE than the MaxDiversity algorithm (proof is in appendix section B). A large $\theta$ suggests a close correlation between the labels of the nodes that are close to each other. It is also common for most of the graph datasets where samples clustered together usually have similar labels. Thus, DiverseUncertainty can achieve a smaller MSE in this case.

**Table 1:** Statistics of graph datasets used in experiments.

| Data | # Nodes | # Train Nods | # Edges | # Classes |
|---|---|---|---|---|
| Cora | 2,708 | 1,208 | 5,429 | 7 |
| Citeseer | 3,327 | 1,827 | 4,732 | 6 |
| Pubmed | 19,717 | 18,217 | 44,328 | 3 |
| Corafull | 19,793 | 18,293 | 126,842 | 70 |
| ogbn-products | 2,449,029 | 196,615 | 61,859,149 | 47 |

# 6 Experiments

We evaluate the performance of *ScatterSample* on five different datasets.

**Datasets.**    We evaluated the different methods on the Cora, Citeseer, Pubmed, Corafull [KW16], and ogbn-products [Hu+20] datasets (Table 1). Besides the ogbn-products, we do not keep original data split of the training and testing set. For the nodes that are not in the validation or testing sets (the validation and testing sets follows the split in the dgl package "dgl.data" [Wan+19]), we will add them to the training set. The labels can only be queried from the training set.

**Baselines.**    For different sampling budget $B$, we compare the test accuracy of *ScatterSample* with the following graph active learning baselines:

- *Random sampling*. Select $B$ nodes uniformly at random from $\mathcal{V}_{train}$.
- *AGE* [CZC17]: AGE computes a score which combines the node centrality, information density, and uncertainty, to select $B$ nodes with the highest scores.
- *ANRMAB* [Gao+18]: ANRMAB learns the combination weights of the three metrics used by AGE with multi-armed bandit method.
- *FeatProp*: FeatProp [Wu+19b] clusters the feature propogations into $B$ clusters and pick the nodes closest to the cluster centers.
- *Grain*: [Zha+21] score the node by the weighted average of the influence score and diversity score. And select the top $B$ nodes with largest node scores. Grain includes two different approaches of selecting nodes, Grain (ball-D) and Grain (NN-D).
- ScatterSample: For the sample scale graph dataset (Cora, Citeseer), we set the initial sampling budget to $3\% \cdot |\mathcal{V}_{train}|$ and sample $1\% \cdot |\mathcal{V}_{train}|$ each round during the dynamic sampling period. For medium scale datasets (Pubmed and Corafull), we set the initial sampling budget to $1\% \cdot |\mathcal{V}_{train}|$ and sample $0.5\% \cdot |\mathcal{V}_{train}|$ each dynamic sampling round. For the large scale dataset (ogbn-products), initial sampling budget is $0.2\% \cdot |\mathcal{V}_{train}|$, and each dynamic sampling round selects $0.05\% \cdot |\mathcal{V}_{train}|$ nodes.

**GNN setup.**    We train a 2-layer GCN network with hidden layer dimension = 64 for Cora, Citeseer and Pubmed, and = 128 for Corafull and obgn-products. To train the GNN, we follow the standard random neighbor sampling where for each node [HYL17], we randomly sample 5 neighbors for the convolution operation in each layer. We use the function in "dgl" package to train the GNNs [Wan+19].

## 6.1   Performance Results

We compare the performance of different active graph neural network learning algorithms under different labeling budgets ($B$). We parameterize the labeling budget $B$ equal to a certain proportion of the nodes in the training set ($B = r|\mathcal{V}_{train}|$). For Cora and Citeseer, we vary $r$ from $5\%$ to $15\%$ in increment of $2\%$; for Pubmed and Corafull, $r$ is varied from $3\%$ to $10\%$; for ogbn-product dataset, we vary the $r$ from $0.3\%$ to $1\%$. The performance of the active learning algorithms are measured with the test accuracy.

**Accuracy.**    Figure 3 shows the test accuracy of baselines trained on different proportions of the selected nodes. *ScatterSample* improves the test accuracy and consistently outperforms other baselines in all the datasets. In Citeseer, *ScatterSample* requires $9\%$ of the node labels to achieve test accuracy $74.2\%$, while the best alternative baselines "Grain (ball-D)" and "Grain (NN-D)" need to label $15\%$ of nodes to achieve similar accuracy, which corresponds to a $40\%$ savings of the labeling cost. Similarly, in PubMed and ogbn-products, *ScatterSample* achieves a $50\%$ labeling cost reduction compared to the best alternative baseline.

**Efficiency.**    Here, we compare the computation time among the methods that use the graph structure and node features to select the samples namely, *ScatterSample*, "Grain (ball-D)" and "Grain (NN-D)". We use the ogbn-products dataset to perform comparisons. *ScatterSample* takes less than 8 hours to determine the labeling nodes and train the GNN, while the Grain algorithm requires more than 240 hours. Grain requires $\mathcal{O}(n^2)$ complexity to calculate the scores of all nodes, which is prohibitive complexity in large graphs.

**Complexity analysis.**    The computation complexity of DiverseUncertainty is $O(|E| + r * B_t^2)$. It is because *ScatterSample* includes two parts: 1) computing the node representations with complexity $O(|E|)$ where $|E|$ is the number of edges and 2) cluster the the uncertain nodes where the complexity is $O(rB_t^2)$. Since both $r$ and $B_t$ are small, $rB_t^2 < |E|$, our method does not add a lot of extra burden compared to the model training time.

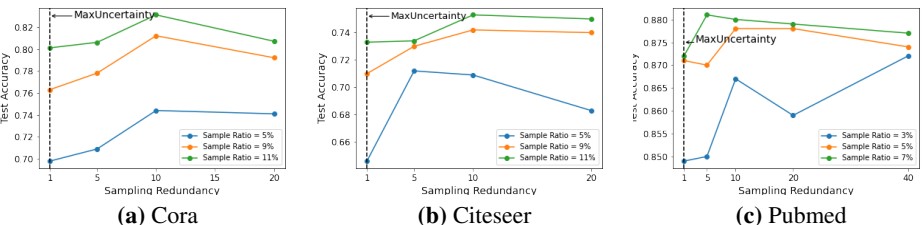

**(a)** Cora      **(b)** Citeseer      **(c)** Pubmed      **(d)** Corafull

**Figure 3:** *ScatterSample* **(blue), wins consistently:** Comparison of the test accuracy of active GNN learning algorithms at different labeling budget. The $x$-axis shows # labeled nodes/# nodes in training set.

## 6.2 Ablation Study

The MaxDiversity algorithm of *ScatterSample* needs to determine the size of candidate set $\mathcal{C}_t$ before selecting a subset $S_t$ from $\mathcal{C}_t$ for labeling. Hence, sampling redundancy $r$ and the clustering algorithm to cluster the nodes in $\mathcal{C}_t$ will affect the performance of *ScatterSample*. In this section, we will evaluate the effect of both factors.

**(a)** Cora      **(b)** Citeseer      **(c)** Pubmed

**Figure 4:** Compare the performance under different sampling redundancy $r$. When $r = 1$, Diverse-Uncertainty reduces to MaxUncertainty method.

**Sampling redundancy $r$:** Recall from algorithm 1, the sampling redundancy $r$ controls the relative size of candidate set $\mathcal{C}_t$ to size of sampled node $S_t$. When $r = 1$, *ScatterSample* reduces to the standard MaxUncertainty algorithm. And figure 4 shows that the sampling the most uncertain nodes is significantly worse than DiverseUncertainty. For the Citeseer dataset, DiverseUncertainty can outperform MaxUncertainty by over 7% when sampling ratio is 5%. Therefore, to achieve a good test accuracy, $r$ should be carefully selected. Figure 4 suggests that as $r$ increases, the test accuracy quickly boosts at the early stage, and then decreases slowly.

**Sensitivity to initial sampling ratio:** During the initial sampling stage, DiverseUncertainty samples $B_0$ nodes to train the model initially. And the initially trained model will affect the nodes sampled during the dynamic sampling period. We test the effect of different initial sampling ratio on Cora and Citeseer datasets. We vary the initial sampling ratio from 2% to 4%, and figure A5 shows that DiverseUncertainty is robust to the choice of initial sampling ratio.

**Diverse uncertainty algorithms:** Besides the sampling algorithm used by DiverseUncertainty, there are some other algorithms to pick the representative nodes from the candidate set $S_t$. First, we will evaluate three algorithms to cluster and select the propagated features.

- Random select: randomly pick nodes $S_t$ from $\mathcal{C}_t$.
- DiverseUncertainty: use $k$-means++ to cluster the nodes in $\mathcal{C}_t$ and
- Random round-robin Algorithm [Cit+21]: use the cluster labels from the initial sampling period (the initial sampling period clusters all the nodes in $\mathcal{V}_{train}$). Then, following the Algorithm A3 (see Appendix) to select $S_t$ from $\mathcal{C}_t$

Figure A6 suggests that $k$-means++ clustering algorithm achieves a better test accuracy in most cases compared to random selection or random round-robin algorithm (see Appendix). Moreover, compared to random sampling algorithm, $k$-means++ clustering algorithm is more robust when the sampling ratio increases. As the sampling ratio increases, the test accuracy of $k$-means++ keeps increasing in most cases, while the test accuracy of random sampling algorithm has more fluctuations.

Another factor that affects the test performance is the metric for clustering. Besides the propagated features (which is used by MaxDiversity), we can also cluster the input features or the embedding

vectors. Since the GNN models typically used do not have a fully connected layer connecting to the output, we cannot use the output of second last layer as the embedding. Hence, we use the GNN output as the embedding vector for clustering. Figure A7 shows that clustering the propagated features consistently outperforms clustering the other two targets. Especially for the "Citeseer" dataset, clustering the propagated features outperforms by at most $5\%$. To conclude, the $k$-means++ clustering algorithm achieves the best performance compared to the other selection methods and clustering the propagated features is better than clustering other targets. Thus, DiverseUncertainty uses $k$-means++ to cluster the propagated features to pick $S_t$ from $\mathcal{C}_t$.

## 7 Empirical validation of theorem

In this section, we perform simulation analysis to demonstrate that *ScatterSample* can reduce the MSE compared to greedy uncertainty sampling approach.

**Graph Simulation Setup.** Let the dimension of input feature $d = 1$. Simulate $\mathbf{X}$ from two different clusters, where $(X|C_1) \sim Uniform(-15, -5)$ and $(X|C_2) \sim Uniform(8, 12)$. In our simulation, we randomly generated 100 nodes for each cluster. Each node is randomly connected to two other nodes in the same cluster. Moreover, for the edges between clusters, we set a probability threshold $r$ such that $\mathrm{P}\left[V_i \in C_1 \text{ connect to a node} \in C_2\right] = r$ (See Appendix D for details).

**Label of nodes.** The label of a node depends on its propagated features. First compute the 1-layer feature propagation of each node, $\mathbf{X}^{(1)}$. Then, the label of $i$-th node is $y_i = |X_i^{(1)}|^2$. Here, because the two cluster centers are equally distanced from 0, hence, the label function is also symmetric around 0.

**Node sampling.** During the initial sampling step, label the nodes closest to the cluster centers and train the GP function. To sample uncertain nodes,

- MaxUncertainty: Label the 8 nodes with largest uncertainty.
- DiverseUncertainty: Collect the top 80 nodes with largest uncertainty into the candidate set. Then, use $k$-means++ to cluster the nodes in the candidate set into 8 clusters. Label the 8 nodes closest to the cluster centers.

MaxUncertainty and DiverseUncertainty use the newly labeled nodes to update the GP function respectively. Finally, the trained GP function predicts the node labels, and we compute the corresponding MSE.

Figure A8 in the Appendix suggests that MaxUncertainty has larger MSE compared to Diverse-Uncertainty algorithm. For the MaxUncertainty algorithm, since most of the labeled nodes come from the cluster 1, the MSE of cluster 1 is significantly smaller than that of cluster 2. While for the DiverseUncertainty algorithm, the MSE of cluster 1 and 2 are comparable. As $r$ increases, there are more and more edges between clusters, and the propagated features are less separated. Hence, there are some high uncertainty nodes from cluster 1 very close to cluster 2, which is beneficial for Max-Uncertainty to learn the labels of nodes from cluster 2. Thus, we could observe $\frac{\text{MSE of MaxUncertainty}}{\text{MSE of DiverseUncertainty}}$ keeps decreasing when $r$ increases. When $r$ is very large, cluster 1 and 2 will merge into one cluster, and MSEs of both methods no longer have a significant difference.

## 8 Conclusion

Learning a GNN model with limited labeling budget is an important but challenging problem. In this paper:

- We propose a novel data efficient GNN learning algorithm, *ScatterSample*, which efficiently diversifies the uncertain nodes and achieves better test accuracy than recent baselines.
- We provide theoretical guarantees: Theorem 5.1 proves the advantage of *ScatterSample* over MaxUncertainty sampling.
- Experiments on real data show that *ScatterSample* can save up to 50% labeling size, for the same test accuracy.

We envision *ScatterSample* will inspire future research of combining uncertainty sampling and representation sampling.

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

## A  Estimation and prediction of the GP model

Given the assumptions and notations above, the likelihood of GP model can be written as:

$$f(\mathbf{y} \mid \mu, \sigma^2, \boldsymbol{\theta}) \propto \exp\left[-\frac{1}{2\sigma^2}(\mathbf{y} - \mathbf{1}\mu)^T \mathbf{K}(\mathbf{X}^{(k)})^{-1}(\mathbf{y} - \mathbf{1}\mu)\right].$$

Here, we assume $\boldsymbol{\theta} = (\theta_1, \theta_2, \dots, \theta_d)$ is a known parameter, and only $\mu$ and $\sigma^2$ are left to fit. The MLE of $\mu$ and $\sigma^2$ are, $\hat{\mu} = \sum_{i=1}^{n} \mathbf{y}_i$ and $\hat{\sigma}^2 = \frac{1}{n}(\mathbf{y} - \hat{\boldsymbol{\mu}})^T \mathbf{K}(\mathbf{X}^{(k)})^{-1}(\mathbf{y} - \hat{\boldsymbol{\mu}})$.

Given a testing point $\mathbf{x}_*^{(k)}$, by the GP model fitted by $D$, the prediction of the response $f(\mathbf{x}_*^{(k)}) \sim N(\mu^*, \sigma^{*2})$, where

$$\mu^* = \mu + k^{*T}\mathbf{K}(\mathbf{X}^{(k)})^{-1}(\mathbf{y} - \mathbf{1}\hat{\mu}) \quad \text{and} \quad \sigma^{*2} = \hat{\sigma}^2(1 - k^{*T}\mathbf{K}(\mathbf{X}^{(k)})k^*) \tag{4}$$
$$k^* = [K(\mathbf{x}_1, \mathbf{x}^*), K(\mathbf{x}_2, \mathbf{x}^*), \dots, K(\mathbf{x}_n, \mathbf{x}^*)]^T \in \mathbb{R}^{n \times 1}$$

## B  Proof of theorem 1

Before proving theorem 5.1, we first provide some preliminary results of Gaussian kernel matrix.

### B.1  Preliminary of Gaussian kernel matrix

**Lemma B.1.** *Let $\mathbf{K}$ be the Gaussian kernel matrix of vector $(\mathbf{c}_1, \mathbf{c}_2, \dots, \mathbf{c}_m)$. Since $\min_{i \neq j} \left\| \mathbf{c}_i - \mathbf{c}_j \right\| > \delta$, we have $\mathbf{K}_{ij} < \exp\left[-\frac{\delta^2}{\theta}\right]$. Denote $\epsilon = \exp\left[-\frac{\delta^2}{\theta}\right]$. Then, $K_{ij}^{-1} > -\epsilon$ if $i \neq j$, and $1 < K_{ii}^{-1} < 1 + (m-1)\epsilon^2$.*

*Proof.* Let $\mathbf{K} = \mathbf{I} + \mathbf{A}$. By Neumann series, $\mathbf{K}^{-1} = \mathbf{I} + \sum_{t=1}^{\infty}(-1)^t \mathbf{A}^t$. Thus, $\mathbf{K}_{ij} > -\mathbf{A}_{ij} > -\epsilon$ for $i \neq j$, and $1 < \mathbf{K}_{ii} < 1 + \mathbf{A}_{ii}^2 < 1 + (m-1)\epsilon^2$ $\qquad\qquad\square$

### B.2  Prove MaxUncertainty method samples $2m$ from cluster $m$

During the initial sampling stage, the nodes at the cluster centers are sampled. Then, the variance of a sample $x$ is,

$$Var(f(x)) = \sigma^2(1 - \mathbf{k}^T \mathbf{K}^{-1} \mathbf{k}), \tag{5}$$

where $\mathbf{k} = (K(x, c_1), K(x, c_2), \dots, K(x, c_m))$ and $\mathbf{K} = \mathbf{K}(\mathbf{c})$ is the Gaussian kernel matrix of $\mathbf{c} = (c_1, c_2, \dots, c_m)$.

For a node $x$ from cluster $i$, the $Var(f(x))$ is monotone increasing as $x$ moves from cluster center to boundary. Let $\omega = \exp\left[-\frac{(\delta - d_m)^2}{2\theta}\right]$. Since $|x - c_j| \geq \delta - d_i \geq \delta - d_m$ for $j \neq i$, naturally, we have $\omega < \mathbf{k}_j$. Then, following lemma B.1,

$$\mathbf{k}(x, \mathbf{c})^T \mathbf{K}^{-1} \mathbf{k}(x, \mathbf{c}) \geq \exp\left[-\frac{d_i^2}{\theta}\right] \mathbf{K}_{ii}^{-1} > \exp\left[-\frac{d_i^2}{\theta}\right] \tag{6}$$

With equation 6, we can upper bound the variance of the $x$ from cluster $i$.

In the next step, we **lower bound the variance of $x$ at the boundary of cluster $m$** (largest cluster), and show that its variance is strictly larger than nodes from other clusters.

$$\mathbf{k}(x, \mathbf{c})^T \mathbf{K}^{-1} \mathbf{k}(x, \mathbf{c}) < \left(\exp\left[-\frac{d_m^2}{\theta}\right] + (m-1)\omega^2\right)[1 + (m-1)\epsilon^2], \tag{7}$$

Since $\delta \geq d_m + \sqrt{d_m^2 + \theta \log(9m)}$, we have $(m-1)\epsilon^2 < (m-1)\omega^2 \leq \frac{1}{9}\exp\left[-\frac{d_m^2}{\theta}\right] < \frac{1}{9}\exp\left[-\frac{d_i^2}{\theta}\right]$.

$$\text{RHS of equation 7} \quad \leq \quad 2\left(\exp\left[-\frac{d_m^2}{\theta}\right] + (m-1)\omega^2\right)$$

$$\leq \quad \frac{1}{2}\exp\left[-\frac{d_i^2}{\theta}\right] + 2(m-1)\omega^2 < \frac{13}{18}\exp\left[-\frac{d_i^2}{\theta}\right] \qquad (8)$$

The RHS of equation 7 is strictly smaller than the LHS of equation 6. Therefore, the uncertainty of nodes at the boundary of cluster $m$ is larger than the uncertainty of nodes from other clusters. For our case, feature dimension is 1 and there only exist 2 points at the boundary of cluster $m$. However, since the nodes are continuous distributed, MaxUncertainty will pick the other $2(m-1)$ nodes close to the boundary of cluster $m$.

### B.3  Bound the MSE of MaxUncertainty and DiverseUncertainty

From previous section, we have seen the boundary nodes of cluster $m$ have the largest uncertainty. Thus, MaxUncertainty will sample $2m$ nodes from the cluster $m$. To lower bound the MSE of MaxUncertainty, we consider the other $(m-1)$ clusters. Since the Gaussian Process model does not have noise, MSE of the prediction is equal to its variance.

Let $\mathbf{h} = (\mathbf{c}, \mathbf{s}) \in \mathbb{R}^{3m}$, where $\mathbf{h}$ are the sampled nodes and $\mathbf{s} \in \mathbb{R}^{2m}$ are the $2m$ nodes sampled during the dynamic sampling stage. Denote $\mathbf{K}(\mathbf{h})$ to be Gaussian kernel matrix of $\mathbf{h}$. Let $t = |x - c_i|$ be the distance from node $x$ to its cluster center.

$$\mathbf{k}(x, \mathbf{h})^T \mathbf{K}^{-1}(\mathbf{h})\mathbf{k}(x, \mathbf{h}) \quad \leq \quad [1 + m\epsilon^2 + 2m\omega^2]\left(\exp\left[-\frac{t^2}{\theta}\right] + 3m\omega^2\right)$$

$$\leq \quad (1 + 3m\omega^2)\left(\exp\left[-\frac{t^2}{\theta}\right] + 3m\omega^2\right) \qquad (9)$$

Moreover, we have $\mathbb{E}_t\left(\exp\left[-\frac{t^2}{\theta}\right]\right) \leq \frac{1}{2}\left(1 + \exp\left[-\frac{d_m^2}{\theta}\right]\right)$. Let $r_i = \exp\left[-\frac{d_i^2}{4\theta}\right]$ and $a = \exp\left[-\frac{d_m^2}{\theta^*}\right]$, we have

$$MSE(f(x) \mid MaxUncertainty, x \in \mathcal{S}_i) > \sigma^2\left[\left(\frac{1}{2} - \frac{a}{2} - \frac{a^2}{9}\right) - \left(\frac{1}{2} + \frac{a}{6}\right)r_i^4\right]. \qquad (10)$$

Hence, $MSE(f(x) \mid MaxUncertainty) > \sigma^2 \sum_{i=1}^{m-1} \frac{d_i^2}{\|\mathbf{d}\|^2}\left[\left(\frac{1}{2} - \frac{a}{2} - \frac{a^2}{9}\right) - \left(\frac{1}{2} + \frac{a}{6}\right)r_i^4\right]$. Let $h(r_i^2) = \left(\frac{1}{2} - \frac{a}{2} - \frac{a^2}{9}\right) - \left(\frac{1}{2} + \frac{a}{6}\right)r_i^4$ and $h$ is concave in $d_i^2$. Thus,

$$h(r_m^2) \quad = \quad \left(\frac{1}{2} - \frac{a}{2} - \frac{a^2}{9}\right) - \left(\frac{1}{2} + \frac{a}{6}\right)r_i^4 \leq \tau \sum_{i=1}^{m-1} \frac{h(r_i^2)}{r_m^2}h(r_m^2) \leq \tau \sum_{i=1}^{m-1} h(r_i^2). \qquad (11)$$

Hence, $MSE(f(x) \mid MaxUncertainty) > \frac{\sigma^2}{1+\tau} \sum_{i=1}^{m} \frac{d_i^2}{\|\mathbf{d}\|^2}\left[\left(\frac{1}{2} - \frac{a}{2} - \frac{a^2}{9}\right) - \left(\frac{1}{2} + \frac{a}{6}\right)r_i^4\right]$.

Then, we upper bound the MSE of DiverseUncertainty. Since each cluster labels 2 nodes at the cluster boundary. For node $x$ from cluster $i$, the distance between node $i$ to the closest labeled point is smaller than $\frac{d_i}{2}$. Hence,

$$\mathbf{k}(x, \mathbf{h})^T \mathbf{K}^{-1}(\mathbf{h})\mathbf{k}(x, \mathbf{h}) \geq \exp\left[-\frac{t^2}{\theta}\right] + \exp\left[-\frac{(d_i - t)^2}{\theta}\right] - 2\exp\left[-\frac{d_i^2}{\theta}\right]\exp\left[-\frac{t^2 + (d_i - t)^2}{2\theta}\right] \geq \frac{2r_i}{1 + r_i^2}. \quad (12)$$

Thus, we have $MSE(f(x) \mid DiverseUncertainty, x \in \mathcal{S}_i) \leq \sigma^2 \frac{(1-r_i)^2}{1+r_i^2}$ and $MSE(f(x) \mid DiverseUncertainty) \leq \sigma^2 \sum_{i=1}^{m} \frac{d_i^2}{\|\mathbf{d}\|^2} \frac{(1-r_i)^2}{1+r_i^2}$.

Moreover, $\frac{MSE(f(x)|MaxUncertainty, x \in \mathcal{S}_i)}{MSE(f(x)|DiverseUncertainty, x \in \mathcal{S}_i)} \geq \frac{1+r_i^2}{1-r_i}\left(\frac{1}{2} + \frac{a}{6}\right) - \frac{(1+r_i^2)}{(1-r_i)^2}\left(\frac{2a}{3} + \frac{a^2}{9}\right)$. Since $\delta \geq d_m + 2\theta \log\left(\frac{3\sqrt{m}}{1-r_m}\right)$, we have $a \leq (1-r_i)^2$ for all $i = 1, 2, \ldots, m$. Thus, $\frac{MSE(f(x)|MaxUncertainty, x \in \mathcal{S}_i)}{MSE(f(x)|DiverseUncertainty, x \in \mathcal{S}_i)} \geq \frac{1}{2}\frac{1+r_i^2}{1-r_i} - \frac{8}{3} \geq \frac{1}{2}\frac{1+r_m^2}{1-r_m} - \frac{8}{3}$.

Now, we could lower bound $\frac{MSE(f(x)|MaxUncertainty)}{MSE(f(x)|DiverseUncertainty)}$ over the whole sample space, where

$$\frac{MSE(f(x) \mid MaxUncertainty)}{MSE(f(x) \mid DiverseUncertainty)} \geq \frac{1}{2(1+\tau)}\frac{1+r_m^2}{1-r_m} - \frac{8}{3}$$

Moreover, when $\theta$ is large, $r_i = \exp\left[-\frac{d_i^2}{4\theta}\right] \approx 1 - \frac{d_i^2}{4\theta}$. Thus, $\frac{MSE(f(x)|MaxUncertainty)}{MSE(f(x)|DiverseUncertainty)} \geq \frac{1}{1+\tau}O(\theta)$.

## C   Ablation Experiments

### C.1   Detail of round-robin algorithm

---

**Algorithm 3** Random Round-robin Algorithm

---

1: Input: cluster labels of node $i$ (node $i \in \mathcal{V}_{train}$) $cl_n$ , where $cl_n \in 1, 2, \ldots, m$; candidate set $\mathcal{C}_t$; number of nodes to label $B_t$.
2: Using the cluster labels to split $\mathcal{C}_t$ onto clusters $A_1, A_2, \ldots, A_m$. Without loss of generality, $|A_1| \leq |A_2| \leq \ldots \leq |A_m|$.
3: $S_t = \emptyset$
4: **for** $i = 1, 2, \ldots, B_t$ **do**
5:    **for** $j = 1, 2, \ldots, m$ **do**
6:      **if** $A_j \neq \emptyset$ **then**
7:       Uniformly select $x$ from $A_j$ at random
8:       $A_j \leftarrow A_j \setminus \{x\}, S_t \leftarrow S_t \cup \{x\}$
9:       break
10:     **end if**
11:   **end for**
12: **end for**
13: return $S_t$

---

### C.2   Sensitivity to initial sampling ratio

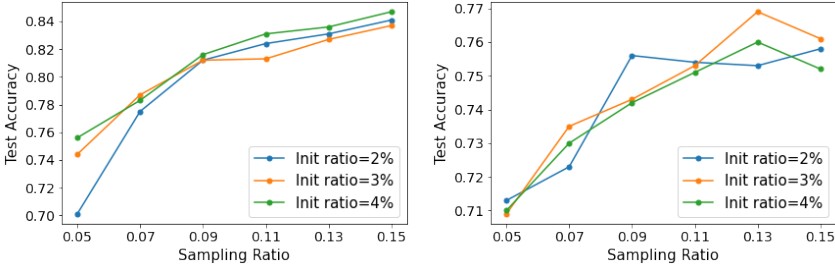

**Figure 5:** Compare different initial sampling ratios for Cora (left) and Citeseer (Right)

## C.3 Compare sampling algorithms

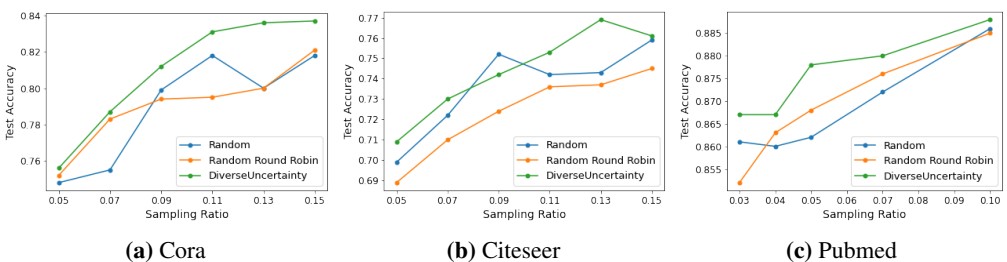

**(a)** Cora      **(b)** Citeseer      **(c)** Pubmed

**Figure 6:** Compare different sampling algorithms to collect $S_t$ from the candidate set $\mathcal{C}_t$.

## C.4 Compare clustering algorithms

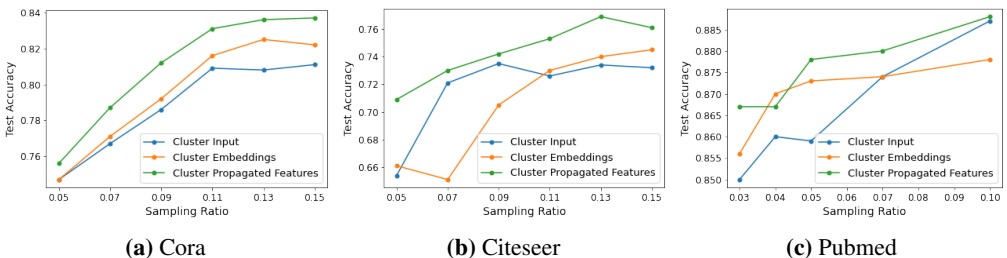

**(a)** Cora      **(b)** Citeseer      **(c)** Pubmed

**Figure 7:** Compare clustering different targets to select $S_t$ from the candidate set $\mathcal{C}_t$.

# D   Empirical validation of theory

**Graph Simulation Setup.** Let the dimension of input feature $d = 1$. Simulate $\mathbf{X}$ from two different clusters, where $(X|C_1) \sim Uniform(-15, -5)$ and $(X|C_2) \sim Uniform(8, 12)$. In our simulation, we randomly generated 100 nodes for each cluster. Then, we simulate the edges between nodes. The edges can be divided into two categories, edges within clusters and edges between clusters. To simulate the edges within clusters, for each node, we random select two other nodes from the same cluster as its neighbor. For the edges between clusters, we set a probability threshold $r$ such that $\mathrm{P}\left[V_i \in C_1 \text{ connect to a node} \in C_2\right] = r$. For each node $V_i \in C_1$, generate an indicator variable $I_i \sim Bernoulli(r)$ to determine whether $V_i$ is connected to cluster 2 ($V_i$ is connected to cluster 2 if $I_i = 1$). If $V_i$ is connected to cluster 2, randomly pick a node from cluster 2 and connect it to $V_i$.

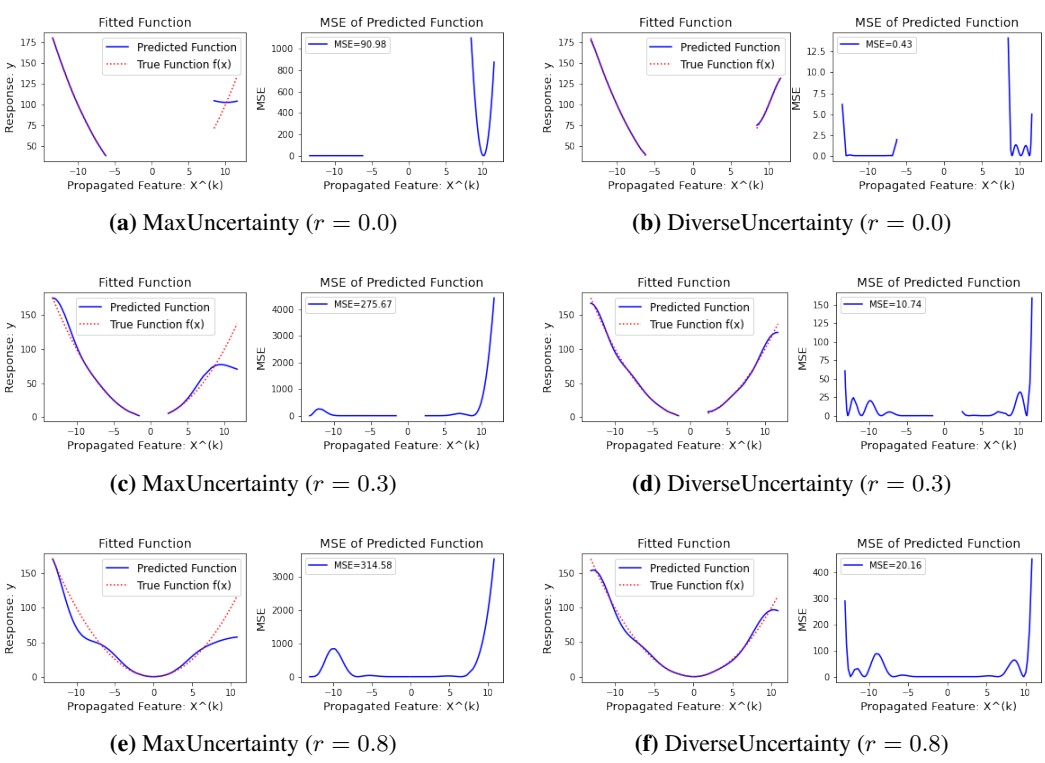

**Figure 8:** Compare the MSEs of Uncertainty and DiverseUncertainty algorithms under different correlation levels between clusters.

