# OpenReview forum: "ScatterSample: Diversified Label Sampling for Data Efficient Graph Neural Network Learning"
_logconference.io/LOG/2022/Conference — LoG 2022 Poster_

### Official Review · Reviewer_w9Rg · 2022-10-21

**Overall Score:** 6
**Confidence:** 3

**Review:**

Contribution:

This paper proposes a data-efficient active sampling framework for training GNNs. The developed sampling mechanism is designed by considering both uncertainty and diversification and thus can lead to smaller test MSE. As a result, the developed ScatterSample method can achieve a comparable test accuracy with other baseline sampling methods with a much smaller sampling ratio.


Strong Points:
* The developed method is effective in saving label collections.
* The paper is well-organized and easy to follow.
* This paper has both theoretical justification and experimental comparison.

Weak points & questions:
* The theoretical analysis is kind of weak: the authors only demonstrate the superior performance of DiverseUncertainty for the GP model. It remains unclear whether the advantage of DiverseUncertainty can be also demonstrated for other models.
* Following the previous comment, is it possible for other models (or just GP model with different hyperparameters), MaxUncertainty can perform better than DiverseUncertainty?

---

### Official Review · Reviewer_Ropm · 2022-10-22

**Overall Score:** 6
**Confidence:** 5

**Review:**

The authors develop an active sampling framework, ScatterSample, to train GNNs under an active learning setting. ScatterSample employs a sampling module termed DiverseUncertainty to collect instances with large uncertainty from different regions of the sample space for labeling. To ensure diversification of the selected nodes, DiverseUncertainty clusters the high uncertainty nodes and selects the representative nodes from each cluster. The proposed ScatterSample algorithm is further supported by rigorous theoretical analysis demonstrating its advantage compared to standard active sampling methods that aim to simply maximize the uncertainty and not diversify the samples. In particular, the authors show that ScatterSample is able to efficiently reduce the model uncertainty over the whole sample space.

Pros: Labeling is an expensive process not only for graph neural networks' training but for all the real-world machine learning tasks. The idea of active learning is the key to this problem. The authors paid their efforts in this research direction and developed the ScatterSample as a solution. The experimental results justify the effectiveness of the authors' work.

Cons: There has been extensive work on the active learning for graph neural networks, such as
"Active learning for node classification: The additional learning ability from unlabelled nodes"
The authors are suggested do a more careful literature review on the related work and elaborate on why the proposed method is better than the existing ones.

---

### Official Review · Reviewer_cW29 · 2022-10-22

**Overall Score:** 6
**Confidence:** 3

**Review:**

This paper proposed a novel data-efficient GNN learning algorithm, which diversifies the uncertain nodes and achieves better test accuracy than recent baselines. The authors also provide theoretical guarantees regarding the advantage of ScatterSample over the MaxUncertainty sampling method. Numerical experiments on real data show that ScatterSample can save up to 50% labeling size, for some given test accuracy.

Originality: This paper proposed the first method that enjoys diversification of the uncertain samples for data-efficient GNN learning. This idea is novel.

Quality: Generally, the theoretical analysis in this paper is technically sound. However, I have some concerns. 1) In Theorem 5.1, why do the authors only consider a case where feature dimension d = 1? It is impractical for applications. 2) The theoretical analysis regarding the advantages of the proposed method is only limited to the MaxUncertainty sampling method. How about other baselines?

Clarity and Significance: The paper is clearly written and could have a high overall impact on the community.

---

### Meta-Review · Area_Chair_QsU3 · 2022-11-17

**Confidence:** 5
**Recommendation:** Accept

**Meta Review:**

This paper proposes a theoretically justified method for data efficient GNN training.  Reviewers appreciated its novelty and the effectiveness of the method as shown in the experimental results.

Strengths:
+ S1.  Relevant problem
+ S2.  Good experimental results
+ S3.  Method's theoretical justification

Weaknesses:
- W1.  Analysis could be improved
- W2.  Some additional work could be cited

In general all reviews are positive on the paper, and it seems like a clear candidate for acceptance.

---

### Decision · Program_Chairs · 2022-11-22

Accept (Poster)